# Psychometric properties of the smartphone addiction scale-short version (SAS-SV) in Honduran university students

Sergio Hidalgo-Fuentes[1,2], Isabel Martínez-Álvarez[3], Fátima Llamas-Salguero[4], Iris Suyapa Pineda-Zelaya[5], César Merino-Soto[6,7], Guillermo M. Chans[6,8]*

1 Department of Psychology and Health, Universidad a Distancia de Madrid (UDIMA), Madrid, Spain, 2 Department of Basic Psychology, Universitat de València, Valencia, Spain, 3 Department of Education, Universidad a Distancia de Madrid (UDIMA), Madrid, Spain, 4 Department of Education Sciences, University of Extremadura, Badajoz, Spain, 5 Francisco Morazán National Pedagogical University, La Ceiba, Atlántida, Honduras, 6 Institute for the Future of Education, Tecnologico de Monterrey, Monterrey, Mexico, 7 Instituto de Investigación FCCTP, University of San Martín de Porres, Lima, Peru, 8 School of Engineering and Sciences, Tecnologico de Monterrey, Mexico City, Mexico

* guillermo.chans@tec.mx

## Abstract

Problematic smartphone use (PSU) is a global issue associated with numerous adverse outcomes, especially among young people. One of the most widely used instruments to evaluate PSU is the Smartphone Addiction Scale-Short Version (SAS-SV). This study examined the psychometric properties of the Spanish version of the SAS-SV, including factorial validity, convergent validity, divergent validity, and reliability. The final sample comprised 530 students from a university in Honduras. Confirmatory factor analysis provided evidence supporting the validity of the instrument's internal structure. Reliability was estimated using McDonald's omega and Cronbach's alpha coefficients. Convergent validity was assessed through correlations with problematic Internet use, depression, anxiety, and stress. Measurement invariance tests were conducted across sex and age categories. The results indicated that the SAS-SV adequately fits a one-dimensional, reliable model and demonstrated measurement equivalence across groups of sex and age. Finally, the SAS-SV demonstrated a strong correlation with problematic Internet use, depression, anxiety, and stress. These findings support the SAS-SV as a valid and reliable instrument for examining PSU among university students in Honduras.

## Introduction

Since the iPhone's launch in 2007, the popularity of smartphones has grown extraordinarily, consolidating as the most widespread digital device in the global market, significantly surpassing other devices such as laptops and tablets [1]. Unlike traditional mobile phones, which were limited to calls and SMS messages, smartphones offer a

**Data availability statement:** The data that support the findings of this study are openly available in Tecnologico de Monterrey Data Hub at https://doi.org/10.57687/FK2/DCVIJU.

**Funding:** The author(s) received no specific funding for this work.

**Competing interests:** The authors have declared that no competing interests exist.

wide range of features, including email, social networking, video games, and internet browsing. It is estimated that there are 4.74 billion smartphone users worldwide [2,3], which continues to grow annually, especially in developing countries where many people access the Internet exclusively through this device [4].

Despite smartphone use's benefits, excessive or uncontrolled use, known as problematic smartphone use (PSU), has been linked to various negative consequences in personal, social, and academic spheres [5–7]. These adverse effects include anxiety [8], depression [9], stress [10], musculoskeletal pain [11], poor sleep quality [12], loneliness [13], academic procrastination [14], poor academic performance [15,16], and generally lower perceived well-being and quality of life [17].

In addition to these various harmful consequences, PSU is highly prevalent, affecting 26.99% of the general population [18], with rates significantly increasing in recent years [19]. This rate is considerably higher in low- or middle-income countries, reaching 43.84% [18]. A recent systematic review by Candussi et al. [20] reveals that PSU is especially prevalent among university students, with rates as high as 67%, underscoring the need for further research in this population.

Although some recent studies have found no difference in PSU prevalence by sex [9,15], numerous others have reported higher rates among women [21–25]. However, it is clear that men and women exhibit different patterns of smartphone use: women tend to focus on social and communication functions, while men are more likely to use smartphones for gaming and Internet browsing [26]. These findings suggest that even if differences in overall PSU levels between men and women are not always observed, distinct usage patterns exist. These observed disparities justify the need for gender-based measurement invariance analysis to determine whether the PSU model is applicable and consistent across both populations.

With respect to age, studies indicate that PSU prevalence varies significantly across age groups, being higher among young adults and university students compared to adolescents and children [18]. These findings highlight age as a relevant factor in understanding differences in problematic smartphone use. The variability in usage patterns across life stages points to the importance of conducting age-based measurement invariance analyses [27], to evaluate whether the factors associated with PSU and the model's structure remain consistent across different age groups. Such analyses provide a more nuanced understanding of how demographic variables influence PSU and offer robust evidence regarding the model's validity across age groups.

Given its high prevalence and associated negative outcomes, it is relevant to discuss how PSU is conceptualized and defined in the scientific literature. Although PSU is often referred to as "smartphone addiction" in the scientific literature due to its similarities to behavioral and substance addictions [28,29], some authors recommend using the term "problematic smartphone use" (PSU). They argue there is insufficient evidence to classify this behavior as an addictive disorder. Additionally, using the term "addiction" can lead to the pathologization and stigmatization of smartphone use [30–32].

PSU is not listed as a disorder in the Diagnostic and Statistical Manual of Mental Disorders (DSM-5) [33] or the International Classification of Diseases (ICD-11) [34]. However, criteria for its diagnosis have been proposed [35,36]. Despite its name, PSU has captured the attention of the scientific community and healthcare professionals due to its serious negative consequences. Studies on this issue have increased significantly recently [37,38], and the World Health Organization considers it a public health problem [39].

The high prevalence of PSU, especially among university students and in low-income countries [40], and its numerous negative consequences make it necessary to have tools for its early detection. A meta-analysis by Harris et al. [41] identified 78 validated scales to assess PSU but concluded that many lacked adequate psychometric properties.

The Smartphone Addiction Scale (SAS) and its short version (SAS-SV) are among the most widely used tests worldwide to evaluate PSU. Developed in South Korea by Kwon et al. [25], the SAS is a multidimensional scale comprising 33 items grouped into six dimensions: disturbance of daily life, positive anticipation, abstinence, cyberspace-oriented relationship, overuse, and tolerance. The items are answered on a six-point Likert scale. The scores of the six dimensions are summed to obtain a total score, where higher scores indicate more PSU. The SAS showed excellent internal consistency in its validation ($\alpha = 0.967$).

The SAS-SV [42] is a short version of the SAS comprising ten items selected by experts. Like the SAS, the items are answered on a six-point Likert scale, with scores ranging from 10 to 60. The authors propose cut-off points to classify excessive smartphone use: 31 for men and 33 for women.

The SAS-SV has a one-dimensional structure and showed high internal consistency in its validation ($\alpha = 0.911$). In addition to its good psychometric properties, the small number of items in the SAS-SV offers several advantages, including shorter application times and higher response rates [43].

These advantages have made the SAS-SV a widely used scale, validated in numerous international samples, including Chinese children and adolescents [44], Brazilian adolescents [45], Iranian adolescents [46,47], Indonesian adolescents [48], Italian adolescents and young people [49], and Moroccan youth and adults [50]. The scale has also been tested on Chinese [21] and American [51] adults. Additionally, studies have been carried out among Turkish [52], Pakistani [53], Serbian [54], Chinese [55], and Italian [22] populations.

Few validations of this instrument have been developed in the Spanish-speaking world. Lopez-Fernandez validated the Spanish adult population scale [56], proposing a cut-off point of 32 to classify subjects with PSU, regardless of sex. Subsequently, Chávez and Rojas-Kramer validated the scale applied to Mexican university students [57].

All validations confirm the SAS-SV's one-dimensional structure, except those performed by Cheung et al. [44] and Zhao et al. [55], who found a better fit with a three-factor structure. The internal consistency of the validations was generally good, with Cronbach's alphas ranging from 0.74 [48] to 0.89 [54,57].

According to the National Telecommunications Commission of Honduras, approximately eight million Hondurans (82.7%) have a mobile phone line. Of these, seven million (71.70% of the population) are mobile Internet subscribers, with most relying on 3G or 4G broadband services [58]. Despite the widespread use of mobile internet, no studies have examined the prevalence or factors associated with PSU in Honduras, revealing a clear need for research in this area.

The rapid growth of smartphone use in Honduras—particularly among younger populations—has raised increasing concerns about digital addiction and its potential impact on mental health. Cultural factors may also significantly contribute to the development of PSU [59].

In addition, the country's economic context and the growing accessibility of mobile technologies have made smartphones an essential part of daily life. However, the potential negative effects on mental health remain insufficiently studied.

These contextual factors underscore the importance of investigating PSU in Honduras, where the sociocultural and technological landscape may differ significantly from that of other regions. Yet, the absence of validated evidence specific to the Honduran population poses challenges to conducting such research. While the Smartphone Addiction Scale–Short Version

 

(SAS-SV) has been translated into Spanish, this alone does not ensure its validity and reliability across all Spanish-speaking contexts, given the considerable geographic and social diversity of the Spanish language [60].

Thus, validating the SAS-SV in the Honduran context is essential, as cultural, socioeconomic, and technological differences may influence how PSU manifests in this population. Such validation would not only address a significant gap in the literature but also establish a stronger foundation for future research on problematic smartphone use across Central America.

This study aimed to analyze the psychometric properties of the SAS-SV scale when applied to Honduran university students. The reliability and validity of the scale in this population were examined. A validated SAS-SV for Honduran university students provides researchers with a reliable instrument to explore the characteristics of PSU in this population group. This information will allow screening evaluations using local or established cut-off points and enable cross-cultural comparisons.

Our hypotheses regarding SAS-SV were as follows:

H1) The scale will demonstrate a unidimensional factor structure.

H2) The scale will demonstrate adequate internal consistency (> 0.80).

H3) The scale will show measurement invariance across sex (male and female) and age groups.

H4) The scale will demonstrate convergent validity through positive correlations with problematic Internet use, depression, anxiety, and stress.

## Method

### Participants

The sample comprised 530 students from the Francisco Morazán National Pedagogical University of Honduras, selected through convenience sampling. Participants ranged in age from 17 to 64 years (M=26.16 and SD=8.33) and were primarily female (Table 1). Following the recommendations of Lloret et al. [61] and Henson et al. [62], who suggest that the exploratory factor analysis (EFA) and confirmatory factor analysis (CFA) should be performed on different samples to avoid misleading conclusions, we divided the total sample into two random subsamples (approximately 50% of the participants each), using the SPSS 28 statistical program.

### Instruments

**Sociodemographic questions.** Sociodemographic data were collected, including participants' age, biological sex, and average number of hours of smartphone use per day.

Table 1. Sociodemographic characteristics of the participants (n=530).

| Variables | M | SD |
|---|---|---|
| Age | 26.16 | 8.33 |
| | N | % |
| Sex | | |
| Male | 124 | 23.4 |
| Female | 406 | 76.6 |
| Hours of smartphone use | | |
| Less than 1 hour per day | 33 | 6.2 |
| Between 1 and 2 hours a day | 86 | 16.2 |
| Between 3 and equi 4 hours a day | 126 | 23.8 |
| More than 4 hours a day | 285 | 53.8 |

Note. M: mean. SD: standard deviation.

**Smartphone addiction scale-short version** [42]. This instrument consists of 10 items scored on a Likert scale from 1 (*Strongly Disagree*) to 6 (*Strongly Agree*). The range of scores is between 10 and 60, where a higher score indicates a higher level of smartphone addiction. This study employed the Spanish adaptation by Lopez Fernandez [56]. As the Honduran author deemed the translation appropriate for the local context, no modifications were made to the original item wording.

**Internet addiction test (IAT)** [63]. Problematic Internet use was assessed using the IAT, which comprises 20 items about the frequency of Internet use behaviors answered on a six-point Likert scale, from 0 (*Never*) to 5 (*Always*). The total score range of the scale is from 0 to 100, with higher scores indicating a higher level of Internet addiction. This study employed the Spanish adaptation by Fernández-Villa et al. [64]. In the present sample, the IAT demonstrated excellent internal consistency, with both McDonald's omega and Cronbach's alpha coefficients of 0.93.

**Depression, anxiety, and stress scale–21 (DASS-21)** [65]. Depression and anxiety were assessed using the DASS-21, a tool designed to measure the negative emotional states of depression, anxiety, and stress. Each was evaluated through a subscale of seven items. Participants indicated their responses regarding their emotional states over the past week using a four-point Likert scale, ranging from 1 (*Does not apply to me at all*) to 4 (*Applies to me very much, or most of the time*). Higher scores reflect higher levels of depression, anxiety, and stress. This study employed the Spanish adaptation by Daza et al. [66]. In our sample, the DASS-21 subscales demonstrated excellent internal consistency: the depression subscale yielded Cronbach's alpha and McDonald's omega coefficients of 0.90; the anxiety subscale, 0.88; and the stress subscale, 0.89.

## Procedure

An online survey was designed using Google Forms to collect the data needed for the study, which was distributed to students via email. The first page of the questionnaire provided information on the objectives and scope of the research, as well as its voluntary and utterly anonymous nature. No personally identifiable information was collected, ensuring participants' anonymity and confidentiality.

Participants were required to provide written informed consent before answering the questions by marking a specific item indicating their understanding. The students did not receive any incentive to participate in the research. The Francisco Morazán National Pedagogical University Research Ethics Committee approved the study, reference number 2023−003.

## Statistical analysis

Given the limited number of studies examining the validity of psychosocial measures in the Honduran population, a sequential analytical strategy was implemented to maximize the extraction of psychometric information. First, responses exhibiting insufficient effort or careless answering were identified to reduce potential sources of response bias (IE/C) [67,68]. Second, item-level descriptive statistics were computed through quantitative analysis. Third, the internal structure of the scale was evaluated in terms of dimensionality, reliability, and measurement invariance across groups. Finally, associations with external variables were examined.

**Potential responses with insufficient effort or carelessness.** Two complementary methods were applied to identify patterns of insufficient effort or carelessness [67,68]: response invariability and response inconsistency [68]. Response invariability was assessed using the longstring method (LS), which counts the number of identical consecutive responses in a participant's answers [69]. Response inconsistency was evaluated using the Mahalanobis distance ($D^2$) method [70]. These two indices—LS and $D^2$—are widely recommended as a minimum screening procedure for detecting IE/C responses [68].

A cutoff of LS > 7 (upper limit based on Tukey's hinge) was used for response invariability. For response inconsistency, participants were flagged if their $D^2$ exceeded the critical chi-square value corresponding to the number of items (i.e.,

$D^2 > \chi^2_{cutoff}$, df: number of items). Participants exceeding both thresholds (LS > 7 and $D^2 > \chi^2_{cutoff}$) were considered to have produced potentially overly consistent or inconsistent responses due to insufficient effort [67–69].

**Item analysis.** To determine the statistical properties, distributional statistics were calculated.

**Internal Structure.** Three types of evidence were examined to evaluate the internal structure of the SAS-SV: dimensionality, reliability, and measurement invariance across groups.

**Modeling.** To determine the most appropriate measurement model for the SAS-SV, dimensionality was assessed using a range of analytic approaches. First, to obtain a baseline of the number of latent dimensions, three methods were applied: empirical Kaiser criterion (EKC) [71], parallel analysis (PA) [72], and the Hull criterion (HULL) [73]. The inter-item correlation matrices were polychoric to ensure consistency with the primary modeling of the SAS-SV.

The CFA was conducted using the ULSMV estimator (Unweighted Least Squares Estimator, with an $\chi^2$ statistic adjusted for mean and variance). This estimator yields more accurate parameter estimates for ordinal data than other estimators [74,75]. The WLSMV estimator (weighted least square mean and variance adjusted) was used for comparison and decision purposes, frequently implemented by treating the items as ordinal categorical variables [76]. After identifying the number of dimensions, the congeneric one-dimensional model (not equality of factor loads) was evaluated. Second, a tau-equivalent model was evaluated, which restricts the loads to equality (a condition typically used for estimating the alpha coefficient). Third, according to the results of Hamamura et al. [77], a structure of three correlated factors was modeled.

**Score reliability.** The omega coefficient [78], appropriate for congeneric models, was calculated; the alpha coefficient was also used in reference to previous studies using this coefficient. An approximation was made to the reliability at different points of the latent attribute. To this end, the items' factor loads and response thresholds (estimated from the results of the inter-item polychoric correlations) were transformed into discrimination and difficulty parameters under the 2-parameter model of the Item Response Theory [78,79].

**Equivalence between groups.** Once the structure was defined, the researchers investigated the equivalence between groups by identifying possible differential functioning between items (DIF). Within the SEM framework, a multiple-indicator-multiple-causes (MIMIC) approach was used [80], where the covariates that may explain DIF (in this study, sex and age) were exogenous variables or predictors of item regression. To test the two forms of DIF, *uniform* ($DIF_{unif}$, DIF constant at the levels of the construct, equivalent to the non-invariance of the intercepts) and *non-uniform* DIF analyses ($DIF_{nunif}$, DIF not constant at the construct levels, equivalent to the non-invariance of factor loads), we employed Restricted Factor Analysis (RFA) [81] with product of indicators (RFA-PI) [82,83]. The PIs represent the interaction between the two covariates. Three predictors of item responses were used: the construct ($\theta$), the grouping variable ($u_1$: sex, $u_2$: age), and the interaction between the construct ($\theta$) and each grouping variable ($\theta u_1$ and $\theta u_2$). The interaction variable is the product $\theta u_i$, doubly centered to maintain orthogonality between predictors. RFA-PI is equivalent to MIMIC modeling and can be interpreted similarly. The parameters for identifying DIF were evaluated with a score test, with $p = 0.05$ corrected with a Bonferroni adjustment [84].

**Association with other variables**. Convergent validity is a form of construct validity evidence that refers to the degree to which a test correlates with other measures that assess theoretically related constructs. Its primary function is to help confirm whether an instrument appropriately measures the intended construct [85].

To assess the convergent validity of the SAS-SV, Pearson's bivariate correlations were calculated between the SAS-SV total score and the following variables: Internet Addiction Test (IAT) scores, DASS-21 subscale scores (depression, anxiety, and stress), and participants' self-reported daily smartphone usage (in hours). The strength of the correlations was interpreted based on the guidelines proposed by Gignac and Szodorai [86], where values around 0.10 are considered small, around 0.20 moderate, and 0.30 or higher are considered strong.

**Prevalence of smartphone users**. Prevalence was estimated based on the cut-off point criteria established by Kwon et al. [42], which initially proposed gender-specific thresholds using receiver operating characteristic (ROC) curve

analysis. However, in the present study, no statistically significant differences were observed between males and females in the total SAS-SV scores. Therefore, a single cut-off point was applied to classify participants as PSU. Differences in total SAS-SV scores between problematic and non-problematic users were analyzed using independent-samples t-tests.

## Results

### Insufficient effort/potential carelessness

Ninety participants were detected by $D^2$, obtaining values equal to or greater than the cut-off point ($D^2 = 23.20$, $p = 0.01$). The longstring method detected 173 subjects at a cut-off point of 7. Due to the low qualifying agreement ($\chi^2 = 21.66$, $p < 0.001$; Cramer's $V = 0.173$, 95% CI = 0.134, 0.198), the cases detected by both methods were removed independently. The final sample was 530.

### Item analysis

The average response of each item tended to be below 3 (*Weakly disagree*), and the response variability was similar (SD < 1.5). The distributional shape of the items was positive asymmetric; all were below 1.5. Regarding kurtosis, all items were below 1.0 (except item 1). See Table 2. In general, there was no excess of asymmetry and kurtosis. However, the absence of multivariate normality (Henze-Zirkler test = 3.81, $p < 0.001$) and univariate normality (Anderson-Darling test between 15.26 and 62.17; all at $p < 0.001$) was detected.

### Internal structure

**Dimensionality.** Using inter-item polychoric correlations, the EKC method yielded the 1st and 2nd eigenvalues at 6.100 and 1.069, respectively; the reference values were 1.294 and 1.151 (percentile method). Similarly, parallel analysis (PA) obtained the 1st and 2nd eigenvalues as 6.100 and 1.069, respectively; the reference values were 1.271 and 1.191 (percentile method). The Hull criterion indicated a single dimension. These results were consistent with item Pearson correlations. In conclusion, EKC, HULL, and PA indicated a latent single dimension.

   **Measurement models fit.** After confirming the unidimensional structure of the SAS-SV, parameter estimates were obtained under two submodels: the congeneric model, which allows item factor loadings to vary, and the tau-equivalent model, which assumes equal factor loadings across all items. Additionally, the multidimensional model proposed by Hamamura et al. [77] was empirically tested to explore alternative factor structures.

**Table 2. Analysis of SAS-SV items.**

|  | M | SD | Sk | Ku | AD |
|---|---|---|---|---|---|
| **SAS1** | 1.81 | 1.20 | 1.46 | 1.46 | 62.17 ** |
| **SAS2** | 2.40 | 1.33 | 0.62 | −0.49 | 24.54 ** |
| **SAS3** | 2.67 | 1.50 | 0.56 | −0.72 | 20.30 ** |
| **SAS4** | 2.49 | 1.47 | 0.77 | −0.41 | 25.52 ** |
| **SAS5** | 2.30 | 1.35 | 0.81 | −0.30 | 29.84 ** |
| **SAS6** | 1.98 | 1.25 | 1.19 | 0.51 | 47.57 ** |
| **SAS7** | 1.88 | 1.22 | 1.29 | 0.81 | 56.30 ** |
| **SAS8** | 2.25 | 1.34 | 0.86 | −0.22 | 31.75 ** |
| **SAS9** | 2.81 | 1.37 | 0.39 | −0.67 | 15.26 ** |
| **SAS10** | 2.06 | 1.32 | 1.23 | 0.79 | 41.67 ** |

Note. M: mean. SD: standard deviation. Min and Max: minimum and maximal response. Sk: skew coefficient. Ku: kurtosis coefficient. AD: normality Anderson-Darling test. **$p < 0.01$.

 

*Congeneric model.* The differences in the adjustment from both estimators did not differ substantially in terms of factor loads and standard errors (Table 3). The fit was acceptable in terms of CFI and SRMR and comparatively better with ULSMV, although it was not adequate in terms of RMSEA. The factor loads with both estimators were high, and their distribution was very similar (ULSMV, M = 0.748, Min = 0.532, Max = 0.902, SD = 0.105; WLSMV, M = 0.761, Min = 0.532, Max = 0.903, SD = 0.108). Due to these results, the ULSMV estimator was chosen. With this estimator, the range of factor loads of the SAS-SV items was between 0.532 and 0.903, which, following Sharma's guidelines [87], identifies factor loads less than 0.32 as poor, those ≥ 0.45 as acceptable, those ≥ 0.55 as good, those ≥ 0.63 as very good, and those above 0.71 as excellent, allowing us to consider that all the SAS-SV items are good indicators of the PSU construct among Honduran university students.

*Tau-equivalent model.* The model was not acceptable compared to the congeneric model due to differences in fit: ULSMV $\chi^2$ = 912.521 (df = 44), CFI = 0.956, SRMR = 0.098, RMSEA = 0.193 (90% CI = 0.182, 0.204).

*Three-factor model.* This model, which comprises three correlated constructs, was tested based on Hamamura's work [77]. ULSMV $\chi^2$ = 70.555 (df = 24), CFI = 0.997, SRMR = 0.031, RMSEA = 0.061 (90% CI = 0.044, 0.077). This model was good in all its fit indicators. However, an examination of local fit (factor loads or interfactor correlations) found very high correlations (Daily~~Withd = 0.720, Daily~~Others = 0.799, Withd~~Others = 0.900). Therefore, the one-dimensional congeneric model was accepted.

**Equivalence between groups.** In sex grouping, tests to identify uniform DIF ($\chi^2$ M = 0.772, Min = 0.023, Max = 1.769) and non-uniform DIF ($\chi^2$ M = 1.088, Min = 0.008, Max = 5.663) were not statistically significant (Table 4). This same result

**Table 3. Estimation of CFA parameters for the SAS-SV.**

| | Factor estimation | | Differences in fitted models | | |
| --- | --- | --- | --- | --- | --- |
| | ULSMV | WLSMV | | | |
| $\chi^2$ (df = 35) | 376.129 | 278.124 | | $\Delta_{fit}$ | |
| **CFI** | 0.983 | 0.987 | | −0.004 | |
| **SRMR** | 0.063 | 0.073 | | −0.010 | |
| **RMSEA** | 0.136 (0.124, 0.148) | 0.115 (0.102, 0.127) | | 0.021 | |
| | | | Factor loadings ($\Delta_F$) | | SE ($\Delta_{se}$) |
| | F | F | $\Delta_{zf}$ | ratio | ratio |
| **SAS1** | 0.698 | 0.713 | −0.030 | 0.979 | 1.124 |
| **SAS2** | 0.644 | 0.670 | −0.046 | 0.961 | 1.048 |
| **SAS3** | 0.532 | 0.532 | 0.000 | 1.000 | 1.078 |
| **SAS4** | 0.734 | 0.796 | −0.150 | 0.922 | 1.061 |
| **SAS5** | 0.840 | 0.893 | −0.215 | 0.941 | 1.076 |
| **SAS6** | 0.902 | 0.903 | −0.005 | 0.999 | 1.147 |
| **SAS7** | 0.820 | 0.823 | −0.009 | 0.996 | 1.134 |
| **SAS8** | 0.785 | 0.780 | 0.013 | 1.006 | 1.122 |
| **SAS9** | 0.761 | 0.751 | 0.023 | 1.013 | 1.121 |
| **SAS10** | 0.769 | 0.756 | 0.031 | 1.017 | 1.052 |

Note. ULSMV: mean- and variance-adjusted unweighted least squares. WLSMV: weighted least squares with mean and variance adjusted. CFI: comparative fit index. SRMR: standardized root mean square. RMSEA: root mean square error of approximation. SAS1 … SAS10: SAS-SV items. SE: standard errors. $\Delta_{fit}$: difference in the statistical test of fit. $\Delta_F$: raw differences in factor loadings. $\Delta_{se}$: ratio of standard errors.

**Table 4. Group equivalence: analysis of differential functioning of SAS-SV items.**

| | Group sex [a] | | Group age [a] | |
|---|---|---|---|---|
| | $DIF_{unif}$ (df = 1) | $DIF_{nunif}$ (df = 1) | $DIF_{unif}$ (df = 1) | $DIF_{nunif}$ (df = 1) |
| SAS1 | 0.675 | 5.663 | 0.087 | 0.671 |
| SAS2 | 0.169 | 0.050 | 1.977 | 0.014 |
| SAS3 | 1.316 | 0.082 | 0.568 | 1.026 |
| SAS4 | 1.769 | 0.372 | 2.714 | 3.381 |
| SAS5 | 0.213 | 0.321 | 0.716 | 0.787 |
| SAS6 | 0.642 | 0.941 | 1.765 | 0.289 |
| SAS7 | 1.496 | 0.012 | 0.000 | 0.038 |
| SAS8 | 0.206 | 0.901 | 0.487 | 1.988 |
| SAS9 | 0.023 | 2.531 | 0.006 | 0.762 |
| SAS10 | 1.283 | 0.008 | 0.731 | 1.526 |

Note. SAS1… SAS10: SAS-SV items. $DIF_{unif}$: $\chi^2$ tests to identify uniform DIF. $DIF_{nunif}$: $\chi^2$ tests to identify non-uniform DIF. df: degrees of freedom. [a] *p*-value of 0.05 with Bonferroni adjustment: 0.05/10 = 0.005. None of the tests were statistically significant.

also occurred in age for uniform DIF ($\chi^2$ M = 0.905, Min = 0.000, Max = 2.714) and non-uniform DIF ($\chi^2$ M = 1.048, Min = 0.14, Max = 3.381). Therefore, the intercepts and factor load of the items were equivalent in these two groups.

**Reliability.** The omega coefficient (0.907; 95% CI = 0.894, 0.920) and the alpha coefficient (0.907; 95% CI = 0.890, 0.918) were high and virtually equal. The approximation of the item information and the score appears in Figs 1 and 2, respectively. The items and the score maintain a consistent level of information in the latent attribute. Specifically, the maximum level of information is attained from the mean to about two SD above the mean. Reliability follows the same pattern (Fig 2).

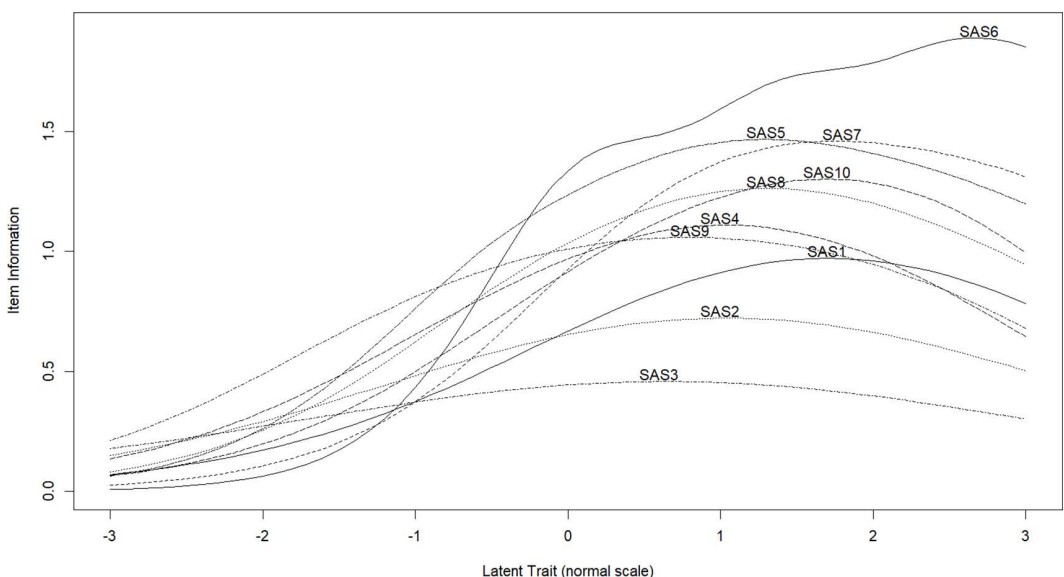

**Fig 1. Information function and reliability for SAS-SV.** Item information from factor analysis. Note: SAS1… SAS10: labels of the SAS-SV items.

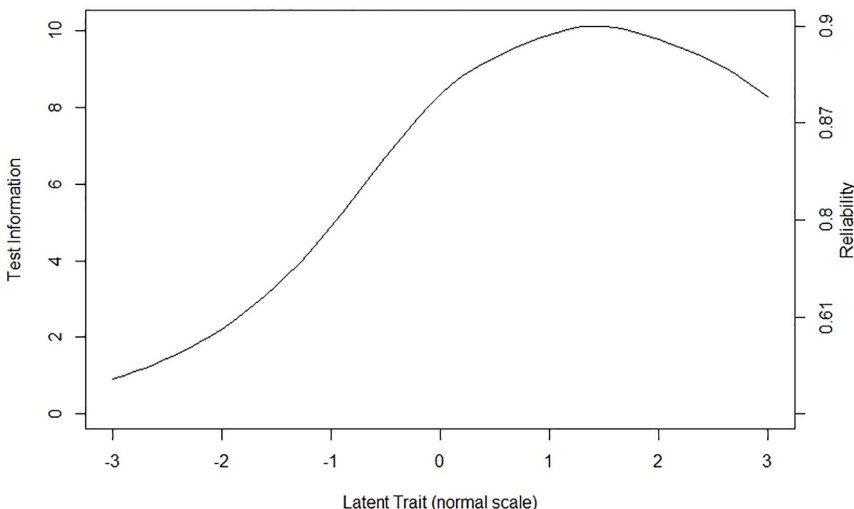

**Fig 2. Information function and reliability for SAS-SV.** Test score information from factor analysis.

## Validity with other variables

Convergent validity was examined through the correlations between the total score of the SAS-SV and the following variables: the total score of the IAT, scores of the DASS21, and the number of hours per day using the smartphone. As shown in Table 5, the total score presented statistically significant positive correlations with problematic Internet use, anxiety, depression, and stress. Following the criterion proposed by Gignac and Szodorai [86], which considers correlations of 0.10, 0.20, and 0.30 to be relatively small, medium, and relatively high, respectively, these correlations are of high magnitude. On the other hand, the correlation between the SAS-SV score and the daily hours of smartphone use was insignificant.

## Prevalence of smartphone users

Kwon et al. [42] originally proposed gender-specific cut-off points for identifying problematic smartphone use (PSU): 31 for males and 33 for females (on a scale ranging from 10 to 60). To estimate the prevalence of PSU in our sample, we followed the procedure used in the Spanish and French adaptations of the SAS-SV [56], beginning with an assessment

**Table 5. Correlation coefficients between the total score of the SAS-SV, IAT, DASS21, and the number of hours per day spent using the smartphone.**

| Variables | r | 95% CI | p |
|---|---|---|---|
| IAT | 0.79 | [0.76; 0.82] | ** |
| DASS21-A | 0.50 | [0.44; 0.55] | ** |
| DASS21-D | 0.49 | [0.43; 0.54] | ** |
| DASS21-S | 0.51 | [0.46; 0.56] | ** |
| Daily hours using the smartphone | 0.01 | [−0.06; 0.08] | |

*Notes.* IAT = Internet Addiction Test; DASS-21-A = Depression, Anxiety, and Stress Scales, anxiety scale; DASS-21-D = Depression, Anxiety, and Stress Scales, depression scale; DASS-21-S = Depression, Anxiety, and Stress Scales, stress scale; *r* = Pearson correlation; 95% CI = Confidence Interval at 95%; *p* = significance level; ** = $p < 0.01$.

of gender differences in total SAS-SV scores. Since no statistically significant differences were found between males and females in our sample, $t(528) = -1.609$, $p = 0.108$, a single average cut-off point of 32 was applied across genders. This approach allowed for a more homogeneous classification of PSU levels among participants.

Using this unified cut-off, 103 students (19.4%) were classified as exhibiting problematic smartphone use. When examined by gender, 18.5% of males and 19.7% of females were classified as PSU users. Statistically significant differences were observed between PSU and non-PSU groups in total SAS-SV scores, $t(528) = -32.775$, $p < 0.001$. Students in the PSU group had a mean score of 39.22 (SD = 6.01), whereas those in the non-PSU group had a mean score of 18.72 (SD = 5.62).

## Discussion

The primary objective of this study was to examine the psychometric properties of the SAS-SV in a sample of Honduran university students. This is the first comprehensive evaluation of the scale's psychometric qualities within the Honduran context. This study offers a preliminary assessment of the SAS-SV's validity and reliability in this specific university population and lays a strong foundation for future research on PSU in Honduras.

An exploratory factor analysis (EFA) was conducted using multiple criteria to determine the optimal number of factors to retain. The results consistently supported a unidimensional structure as the most appropriate for the SAS-SV. Subsequently, several factor models were tested through confirmatory factor analysis. The tau-equivalent unidimensional model demonstrated poor fit and was therefore deemed inadequate. In contrast, the three-factor model proposed by Hamamura et al. [77], showed acceptable global fit indices; however, the extremely high correlations among the three factors indicated a lack of discriminant validity among the dimensions. Based on these findings, the congeneric unidimensional model was selected as the most suitable representation of the scale.

As an additional note, the results for the RMSEA were inconsistent with our conclusion regarding the goodness of fit of the SAS-SV measurement model. However, the inclusion of RMSEA to conclude about the fit of SEM models, although common, is controversial due to the influence of the model's degrees of freedom [88,89]. Importantly, all other fit indices supported the conclusion of adequate model fit, thereby providing continued support for the selected measurement model.

The internal consistency of the SAS-SV was analyzed using Cronbach's alpha and McDonald's omega coefficients, both obtaining values of 0.91. According to the criteria established by Nunnally et al. [90], these values indicate that the scale is suitable for both research and clinical applications. The scale is intended to facilitate the early identification of students with PSU, enabling timely interventions to mitigate this behavior and prevent its associated negative consequences.

Regarding the Spanish-language adaptations, reliability coefficients have also been consistently high across different linguistic and cultural contexts. The validation conducted in a Spanish population reported a Cronbach's alpha of 0.88 [56], while the Mexican adaptation showed an alpha of 0.89 [57]. These findings suggest that the Spanish versions of the SAS-SV maintain strong internal consistency across Spanish-speaking populations, reinforcing the robustness and cross-cultural applicability of the instrument.

Regarding convergent validity, it was hypothesized that the SAS-SV score would present positive correlations with problematic Internet use, depression, anxiety, and stress. The SAS-SV score in our study showed a highly positive correlation with the IAT score. Although it is essential to distinguish between PSU and problematic Internet use, as differences have been found in the prevalence and predictor variables of both phenomena [18,23,31,91]), several studies have found overlap between the two constructs [92–94] beyond the use of Internet-based technology. Likewise, a recent meta-analysis found neuroanatomical similarities between adolescents and young people with PSU and problematic Internet use, mainly pertaining to executive functions and reward processing [95]. Besides the similarities between the PSU and problematic Internet use, it is noteworthy that the IAT dimensions (excessive use, neglect of social life and work, anticipation, salience, lack of control) are similar to the symptoms assessed by the SAS-SV [45].

The total score of the SAS-SV also showed high positive correlations with the three DASS-21 scales, consistent with the results of various meta-analyses and reviews that found PSU associated with psychological distress, especially anxiety and depression [7,96–98]. Although the direction of this correlation remains controversial, some studies claim that PSU increases levels of anxiety and depression [99–101], while others found that psychological distress increases the risk of developing PSU [102–105]. In line with the latter, the theoretical model proposed by Billieux and Maurage [30] argues that one of the pathways that can lead to developing PSU is the "reassurance pathway," referring to individuals driven to use their smartphones by the need to maintain relationships for the security of having others; their risk factors would include a high level of anxiety. Integrating both proposals, some studies have found that the correlation between PSU and psychological distress is bidirectional [106,107].

Finally, although several studies have found a positive correlation between the SAS-SV score and self-reported smartphone usage time [45,56,108–110], the present study found no statistically significant correlation between these variables. It should be noted that PSU and smartphone usage time are not the same construct, and high smartphone usage time alone does not necessarily cause adverse effects [19]. Additionally, self-reported time spent using digital devices, like smartphones, should not be taken as a central explanatory variable for the problematic use of new technologies because it is not an objective reflection of their actual use [111].

The analysis of DIF, including both uniform and non-uniform DIF, revealed no statistically significant differences in item performance across sex and age groups. Specifically, the $\chi^2$ values for both types of DIF did not reach statistical significance, suggesting that the test items function equivalently and consistently across the subgroups evaluated. This finding is particularly important as it reinforces the measurement invariance of the instrument across diverse populations.

The absence of DIF implies that the scale measures the same underlying construct regardless of sex or age, thereby reducing the risk of bias in the results. This evidence of measurement invariance strengthens the instrument's construct validity, indicating that demographic variables do not systematically influence participants' responses. Such invariance is essential when generalizing findings across different subpopulations and supports the use of the instrument in diverse research settings without concern that sex- or age-related bias will compromise the interpretation of results.

With regard to the prevalence of PSU in our sample, which was 19.4%, this figure differs from the 26.96% reported in a recent meta-analysis of university students [18]. Several factors may account for this discrepancy, including the specific characteristics of our sample, the cultural context in which the study was conducted, and the methodological approaches employed. Despite this variability, our findings support the notion that PSU is a relevant concern in university settings, even though its prevalence may vary depending on the characteristics of the population studied.

This study has several limitations that should be considered when interpreting its findings. First, participants were recruited through convenience sampling from a single public university focused exclusively on teacher training. This institution primarily serves students from Indigenous communities and has one of the lowest tuition costs among universities in Honduras. These factors may influence participants' socioeconomic background and access to technology, thereby limiting the generalizability of the results. Future research in Honduras should aim to include more diverse and representative samples to confirm the present findings.

Second, the study relied on self-report measures, which may be subject to response biases such as social desirability. This limitation could affect the validity of the data. Future research could benefit from the use of passive monitoring techniques to objectively assess smartphone use. These methods allow for the collection of accurate and ecologically valid data (e.g., screen time) without requiring active engagement from participants [112].

Third, although participants' smartphone usage time was recorded, the specific ways in which they use this technology were not examined. Future research should investigate the particular smartphone activities university students engage in to better understand their impact on PSU.

Lastly, this study did not evaluate the test–retest reliability of the SAS-SV. Future studies should address this limitation by examining the scale's temporal stability within the Honduran population to ensure the consistency of scores over time in this context.

Despite these limitations, the study includes a notable strength: the evaluation of response bias due to insufficient effort or careless responding (IE/C). This response pattern has been discussed in the literature for several years [69] and has more recently been recognized as a significant source of error that can compromise the validity of results in both psychometric and non-psychometric research relying on self-reports [67–69]. With the exception of a few recent studies conducted in Spanish-speaking contexts [113–115], methods for detecting IE/C responses remain infrequently used in empirical research with Spanish-speaking samples.

In the present study, although the detection of IE/C resulted in the exclusion of approximately one-third of the original sample, this decision contributed to two important outcomes: (1) it enhanced the internal validity of the findings by minimizing potential distortions, and (2) the proportion of excluded cases was consistent with prevalence rates reported in the literature [116,117].

## Conclusions

Despite the aforementioned limitations, this study provides robust evidence supporting the psychometric validity of the Spanish version of the SAS-SV among Honduran university students. The scale demonstrated strong factorial validity, high internal consistency, and measurement invariance across sex and age groups. Furthermore, the strong positive correlations between SAS-SV scores and indicators of psychological distress—such as depression, anxiety, and stress—highlight the relevance of addressing problematic smartphone use (PSU) within the broader context of mental health, given its close association with emotional well-being.

These findings suggest that the SAS-SV can serve as an effective early screening tool for identifying students at risk of PSU, offering valuable insights into the relationship between smartphone use and mental health. Owing to its psychometric robustness, the scale may be integrated into mental health screenings within educational institutions to facilitate the early detection of at-risk individuals and support timely intervention. Early identification and appropriate support may help prevent the escalation of PSU into more severe psychological conditions, such as chronic anxiety, depression, or stress-related disorders.

In conclusion, the SAS-SV emerges as a valuable tool for both research and practice, enabling the early identification of PSU and supporting intervention strategies aimed at mitigating its negative psychological effects. Given the increasing prevalence of smartphone use among young adults—particularly in academic settings—the implementation of validated screening tools such as the SAS-SV may carry significant public health benefits. Their use could promote healthier digital habits and contribute to the improvement of overall mental health in the population.

## Acknowledgments

The authors thank the participants for their involvement in responding to the survey. We also acknowledge the financial support provided by the Writing Lab, Institute for the Future of Education, Tecnologico de Monterrey, Mexico, which was instrumental in developing this work.

## Author contributions

**Conceptualization:** Sergio Hidalgo-Fuentes, Isabel Martínez-Álvarez, Fátima Llamas-Salguero.

**Data curation:** Sergio Hidalgo-Fuentes.

**Formal analysis:** Sergio Hidalgo-Fuentes, César Merino-Soto.

**Funding acquisition:** Guillermo M. Chans.

**Investigation:** Sergio Hidalgo-Fuentes, Iris Suyapa Pineda-Zelaya.

**Methodology:** César Merino-Soto.

**Project administration:** Sergio Hidalgo-Fuentes, Isabel Martínez-Álvarez, Fátima Llamas-Salguero.

**Resources:** Iris Suyapa Pineda-Zelaya.

**Supervision:** Sergio Hidalgo-Fuentes, Isabel Martínez-Álvarez, Fátima Llamas-Salguero.

**Visualization:** Sergio Hidalgo-Fuentes.

**Writing – original draft:** Sergio Hidalgo-Fuentes, César Merino-Soto.

**Writing – review & editing:** Sergio Hidalgo-Fuentes, César Merino-Soto, Guillermo M. Chans.

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
