## [Decision Letter · Decision Letter 0]

26 Feb 2025

PONE-D-24-42743
Psychometric properties of the smartphone addiction scale (SAS-SV) in Honduran university students
PLOS ONE

Dear Dr. Chans,

Thank you for submitting your manuscript to PLOS ONE. After careful consideration, we feel that it has merit but does not fully meet PLOS ONE’s publication criteria as it currently stands. Therefore, we invite you to submit a revised version of the manuscript that addresses the points raised during the review process. Please carefully address reviewers' suggestions and queries. 

We look forward to receiving your revised manuscript.

Kind regards,

Marianna Mazza

Academic Editor

PLOS ONE

“The authors thank the participants for their involvement in responding to the survey.

The authors acknowledge the financial and technical support of the Writing Lab, Institute for the Future of Education, Tecnologico de Monterrey, Mexico, in producing this work.”

“The author(s) received no specific funding for this work.’

Reviewers' comments:

Reviewer's Responses to Questions

**Comments to the Author**

1. Is the manuscript technically sound, and do the data support the conclusions?

Reviewer #1: Yes

Reviewer #2: Yes

2. Has the statistical analysis been performed appropriately and rigorously? 

Reviewer #1: Yes

Reviewer #2: Yes

3. Have the authors made all data underlying the findings in their manuscript fully available?

Reviewer #1: Yes

Reviewer #2: Yes

4. Is the manuscript presented in an intelligible fashion and written in standard English?

Reviewer #1: No

Reviewer #2: Yes

5. Review Comments to the Author

Reviewer #1: Some of the language needs to be proofread and reorganized to facilitate readers' understanding.

In addition, during the data screening process, it was detected that one-third of the data was invalid data, and the valid ratio of the subjects' responses was relatively low, which may have a potential impact on the representativeness of the sample and the robustness of the data results. This issue needs to be treated with caution.

The specific suggestions are provided in the Recommendation document.

Reviewer #2: The study provides valuable insights into the psychometric properties of the SAS-SV in a Honduran university student population. This is a useful contribution, given the limited validation studies in Spanish-speaking countries.

The manuscript is well-written and generally clear. However, some sections could benefit from further clarification and detail.

Specific Remarks:

Abstract:

To enhance the abstract, we recommend incorporating key details aligned with the IMRAC structure (Introduction , Method , results , and conclusion) and the journal's specific guidelines, ensuring a comprehensive and informative summary of the study

Introduction:

The introduction effectively highlights the importance of studying PSU and the need for validated assessment tools. However, it could be strengthened by:

Adding a sentence or two about the specific context of smartphone use in Honduras or Latin America. Are there any unique factors that make this population particularly relevant to study?

Briefly mentioning the cultural adaptation process of the SAS-SV. Was any adaptation beyond translation necessary?

Methods:

Participants:

The convenience sampling method is acknowledged as a limitation, but more discussion on the potential biases this may introduce would be beneficial. For example, are students from this particular university likely to be different from other Honduran students in terms of socioeconomic status or access to technology?

Clarify whether participants received course credit or any other form of compensation for their participation.

Instruments:

For the SAS-SV, provide more detail on the specific Spanish adaptation used (e.g., who translated it, what changes were made, and whether it has been used in other studies).

For the DASS-21, mention the specific instructions given to participants regarding the time frame they should consider when answering the questions (e.g., "In the past week...").

Procedure:

Specify how the online survey was distributed to students (e.g., through university email lists, social media, etc.).

Elaborate on the measures taken to ensure anonymity and confidentiality of the data.

Analysis:

Explain the rationale for using the ULSMV estimator in the confirmatory factor analysis. Why was this estimator chosen over other alternatives?

Provide more details on the methods used to detect and handle careless responses (e.g., what specific criteria were used to identify participants with insufficient effort?).

Results:

When reporting the CFA results, include the values of other relevant fit indices (e.g., CFI, TLI, RMSEA) in addition to the chi-square statistic.

In the invariance testing section, report the change in CFI or RMSEA values used to determine whether invariance was supported.

Discussion:

The discussion adequately addresses the limitations of the study. However, it could be expanded to:

Discuss the implications of the findings for researchers and practitioners working with Honduran university students. How can the SAS-SV be used to identify and support students with PSU?

Suggest specific directions for future research, such as exploring the relationship between PSU and academic performance or mental health outcomes in this population.

Clinical Utility: Discuss the potential clinical utility of the SAS-SV for identifying students who may be at risk for problematic smartphone use and for developing interventions to address this issue.

Tables:

Ensure that all tables are clearly labeled and that all abbreviations are defined in the table footnotes.

References:

Check the accuracy and completeness of all references.

6. PLOS authors have the option to publish the peer review history of their article (what does this mean?). If published, this will include your full peer review and any attached files.

Reviewer #1: No

Reviewer #2: **Yes: **Mohamed Amine BABA

---

## [Author Response · Author response to Decision Letter 1]

7 Apr 2025

Editor

1. Thank you very much for the reminder. We have reviewed the manuscript to ensure it meets all the journal's requirements.

2. Thank you for your feedback. As requested, we have removed the funding-related text from the Acknowledgments section and have updated our Funding Statement accordingly.

3. We have uploaded the dataset to the Tecnológico de Monterrey’s Data Hub, which is public. We added the following text:

Data availability statement

“The data that support the findings of this study are openly available in Tecnologico de Monterrey Data Hub at https://doi.org/10.57687/FK2/DCVIJU.”

4. We have not included any supporting information files. All citations are up to date.

Reviewer #1

1. Thank you for your valuable feedback. We have restructured the abstract to provide a clearer and more detailed description of the research methods. Specifically, we have explicitly stated the types of validity assessed and clarified the variables used to evaluate these relationships (problematic Internet use, depression, anxiety, stress, and resilience). Additionally, we have ensured that the information is more specific and transparent to enhance the reader’s understanding.

2. Thank you for your insightful feedback. We have revised the sentence to clarify that the SAS-SV demonstrated strong correlations with problematic Internet use, depression, anxiety, and stress. We have removed the abbreviations (IAT and DASS-21) from this section to ensure better clarity and avoid confusion for readers unfamiliar with these terms.

3. Thank you for your suggestion. We have added a new source to complement our reference list. However, to the best of our knowledge, there is currently no official government report or peer-reviewed scientific article that provides data on the global number of smartphone users. That said, both the Statista report and the source we used are widely cited in numerous peer-reviewed articles, which reinforces their credibility. Regarding data specific to Honduras, we refer to government sources such as the National Telecommunications Commission of Honduras, which reports that approximately eight million Hondurans (82.7%) have a mobile phone line, and seven million (71.7%) are mobile Internet subscribers [55]. We appreciate your feedback and will continue to prioritize authoritative sources when available.

4. Thank you for your valuable feedback. We have expanded the introduction to better justify the need for analyzing measurement invariance across sex and age. In the revised version, we provide a clearer rationale for the inclusion of these analyses, addressing the relevance of gender and age as demographic factors in understanding the patterns of problematic smartphone use (PSU). Specifically, we now highlight the inconsistent findings in the literature regarding gender differences in PSU and the significant variability in PSU prevalence across different age groups. This additional explanation underscores the importance of testing for measurement invariance to ensure the applicability and validity of the PSU model across different subpopulations.

Line 67:

“However, it is clear that men and women exhibit different patterns of smartphone use: women tend to focus on social and communication functions, while men are more likely to use smartphones for gaming and Internet browsing [26]. These findings suggest that even if differences in overall PSU levels between men and women are not always observed, distinct usage patterns exist. These observed disparities justify the need for gender-based measurement invariance analysis to determine whether the PSU model is applicable and consistent across both populations.

 With respect to age, studies indicate that PSU prevalence varies significantly across age groups, being higher among young adults and university students compared to adolescents and children [18]. These findings highlight age as a relevant factor in understanding differences in problematic smartphone use. The variability in usage patterns across life stages points to the importance of conducting age-based measurement invariance analyses [27], to evaluate whether the factors associated with PSU and the model’s structure remain consistent across different age groups. Such analyses provide a more nuanced understanding of how demographic variables influence PSU and offer robust evidence regarding the model’s validity across age groups.

Line 498

“The analysis of DIF, including both uniform and non-uniform DIF, revealed no statistically significant differences in item performance across sex and age groups. Specifically, the χ2 values for both types of DIF did not reach statistical significance, suggesting that the test items function equivalently and consistently across the subgroups evaluated. This finding is particularly important as it reinforces the measurement invariance of the instrument across diverse populations.

The absence of DIF implies that the scale measures the same underlying construct regardless of sex or age, thereby reducing the risk of bias in the results. This evidence of measurement invariance strengthens the instrument’s construct validity, indicating that demographic variables do not systematically influence participants’ responses. Such invariance is essential when generalizing findings across different subpopulations and supports the use of the instrument in diverse research settings without concern that sex- or age-related bias will compromise the interpretation of results.”

5. We have revised the text to enhance the logical connection between the paragraph discussing the prevalence of PSU and the one addressing the terminology used to refer to the phenomenon. Specifically, we added the following sentence to bridge the two sections:

Line 92

"Given its high prevalence and associated negative outcomes, it is relevant to discuss how PSU is conceptualized and defined in the scientific literature."

This addition helps create a clearer flow between the prevalence of PSU and the ongoing debate over its definition, improving the overall coherence of the paragraphs.

6. Thank you for your insightful comment. In response, we have added a section highlighting the specific need for validating the Smartphone Addiction Scale-Short Version (SAS-SV) in the Honduran context. While the SAS-SV has been translated into Spanish, we emphasize that this does not guarantee its validity and reliability across all Spanish-speaking contexts due to the geographic and social variability of the Spanish language. Therefore, validating the scale in the Honduran population is crucial, considering the cultural, socioeconomic, and technological differences that may influence the manifestation of problematic smartphone use in this specific population.

Line 142

“The rapid growth of smartphone use in Honduras—particularly among younger populations—has raised increasing concerns about digital addiction and its potential impact on mental health. Cultural factors may also significantly contribute to the development of PSU[59].

In addition, the country's economic context and the growing accessibility of mobile technologies have made smartphones an essential part of daily life. However, the potential negative effects on mental health remain insufficiently studied.

These contextual factors underscore the importance of investigating PSU in Honduras, where the sociocultural and technological landscape may differ significantly from that of other regions. Yet, the absence of validated evidence specific to the Honduran population poses challenges to conducting such research. While the Smartphone Addiction Scale–Short Version (SAS-SV) has been translated into Spanish, this alone does not ensure its validity and reliability across all Spanish-speaking contexts, given the considerable geographic and social diversity of the Spanish language [60].

Thus, validating the SAS-SV in the Honduran context is essential, as cultural, socioeconomic, and technological differences may influence how PSU manifests in this population. Such validation would not only address a significant gap in the literature but also establish a stronger foundation for future research on problematic smartphone use across Central America.”

7. Thank you for your valuable feedback. We understand your recommendation to present the research hypotheses in the order of reliability, validity, questionnaire structure, and measurement invariance. However, based on the logical sequence of validity evidence in psychometric research, we have opted for a slightly different order that we believe better reflects the flow of evidence collection: 1) Dimensionality, where we assess the structure of the construct to ensure the instrument measures what it is intended to measure; 2) Reliability, where we evaluate the internal consistency of the scale; 3) Equivalence between groups, ensuring the instrument works consistently across different groups (gender groups and age); and 4) Convergent Validity, where we correlate our scale with other established measures of related constructs. We believe this sequence provides a more logical and natural progression in presenting validity evidence, and we hope this clarifies our approach. We appreciate your understanding.

8. We have updated the Introduction section based on your feedback. We focused more on the main topic, included further evidence to highlight the importance of the research, and improved the overall flow and coherence of the content.

9. Thank you for your observation. We have clarified the sampling technique in the revised manuscript.

10. Thank you for bringing this to our attention. We have now included the reliability coefficients for each instrument used in the study, including the Cronbach's alpha and McDonald's omega values, to ensure clarity on the internal consistency of the scales employed.

11. We have made the necessary revisions to the Statistical Analysis section, adjusting the labeling and providing a clearer explanation of the analysis process. We also included an introductory overview of the methods used.

12. In the discussion, we mentioned that the exclusion of participants due to insufficient effort or potential carelessness actually strengthens the validity of the study. By removing those who were not fully engaged, we ensure that the data more accurately reflects the target sample—those who actively and seriously participated in the study. This data cleaning process improves the quality of the results and ensures that the analyses are based on more reliable data, which in turn strengthens the internal validity of the study.

Line 541

“Despite these limitations, the study includes a notable strength: the evaluation of response bias due to insufficient effort or careless responding (IE/C). This response pattern has been discussed in the literature for several years [69], and has more recently been recognized as a significant source of error that can compromise the validity of results in both psychometric and non-psychometric research relying on self-reports [67-69]. With the exception of a few recent studies conducted in Spanish-speaking contexts [113-115], methods for detecting IE/C responses remain infrequently used in empirical research with Spanish-speaking samples.

In the present study, although the detection of IE/C resulted in the exclusion of approximately one-third of the original sample, this decision contributed to two important outcomes: (1) it enhanced the internal validity of the findings by minimizing potential distortions, and (2) the proportion of excluded cases was consistent with prevalence rates reported in the literature [116, 117].”

13. The results have been organized according to the order of the research hypotheses, ensuring a clear and structured presentation of the findings. Thank you for your suggestion.

14. The order of presentation follows the standard format commonly used in validations, ensuring a clear and structured presentation of the results.

15. Thank you for your valuable feedback. In response to your suggestion, we have expanded the discussion section to provide a more comprehensive and in-depth analysis of the research results.

Line 446

“As an additional note, the results for the RMSEA were inconsistent with our conclusion regarding the goodness of fit of the SAS-SV measurement model. However, the inclusion of RMSEA to conclude about the fit of SEM models, although common, is controversial due to the influence of the model’s degrees of freedom [88, 89]. Importantly, all other fit indices supported the conclusion of adequate model fit, thereby providing continued support for the selected measurement model.”

16. Thank you for your comment. We have followed the procedure used in the validation of the Spanish version of the SAS-SV, which we used to classify the participants. This approach ensures that the classification of smartphone addiction in our study aligns with the established methodology, providing a clear understanding of the participants' status regarding problematic smartphone use. We appreciate your feedback, as it helps clarify these important aspects of the study.

Line 414

“Kwon et al. [42] originally proposed gender-specific cut-off points for identifying problematic smartphone use (PSU): 31 for males and 33 for females (on a scale ranging from 10 to 60). To estimate the prevalence of PSU in our sample, we followed the procedure used in the Spanish and French adaptations of the SAS-SV [56], beginning with an assessment of gender differences in total SAS-SV scores. Since no statistically significant differences were found between males and females in our sample, t(528) = −1.609, p = 0.108, a single average cut-off point of 32 was applied across genders. This approach allowed for a more homogeneous classification of PSU levels among participants.

Using this unified cut-off, 103 students (19.4%) were classified as exhibiting problematic smartphone use. When examined by gender, 18.5% of males and 19.7% of females were classified as PSU users. Statistically significant differences were observed between PSU and non-PSU groups in total SAS-SV scores, t(528) = −32.775, p < .001. Students in the PSU group had a mean score of 39.22 (SD = 6.01), whereas those in the non-PSU group had a mean score of 18.72 (SD = 5.62).”

17. Thank you for your comment. In response, we have added a more detailed discussion regarding the results of measurement invariance across sex and age groups. This additional explanation addresses the implications of these findings and explores potential reasons for the lack of significant differences in the functioning of the test items across these groups. We believe this addition enhances the clarity and depth of the discussion on this important aspect of the study.

Line 498

“The analysis of DIF, including both uniform and non-uniform DIF, revealed no statistically significant differences in item performance across sex and age groups. Specifically, the χ2 values for both types of DIF did not reach statistical significance, suggesting that the test items function equivalently and consistently across the subgroups evaluated. This finding is particularly important as it reinforces the measurement invariance of the instrument across diverse populations.

The absence of DIF implies that the scale measures the same underlying construct regardless of sex or age, thereby reducing the risk of bias in the results. This evidence of measurement invariance strengthens the instrument’s construct validity, indicating that demographic variables do not systematically influence participants’ responses. Such invariance is essential when generalizing findings across different subpopulations and supports the use of the instrument in diverse research settings without concern that sex- or age-related bias will compromise the interpretation of results.”

18. Thank you for your valuable feedback. In response to your recommendation, we have expanded the discussion section to explicitly highlight the contributions of our research and provide suggestions for future research.

Line 563

“These findings suggest that the SAS-SV can serve as an effective early screening tool for identifying students at risk of PSU, offering valuable insights into the relationship between smartphone use and mental h

---

## [Decision Letter · Decision Letter 1]

12 Jun 2025

Psychometric properties of the smartphone addiction scale (SAS-SV) in Honduran university students

PONE-D-24-42743R1

Dear Dr. Chans,

We’re pleased to inform you that your manuscript has been judged scientifically suitable for publication and will be formally accepted for publication once it meets all outstanding technical requirements.

Kind regards,

Frantisek Sudzina

Academic Editor

PLOS ONE

Additional Editor Comments (optional):

Reviewers' comments:

Reviewer's Responses to Questions

**Comments to the Author**

1. If the authors have adequately addressed your comments raised in a previous round of review and you feel that this manuscript is now acceptable for publication, you may indicate that here to bypass the “Comments to the Author” section, enter your conflict of interest statement in the “Confidential to Editor” section, and submit your "Accept" recommendation.

Reviewer #2: All comments have been addressed

2. Is the manuscript technically sound, and do the data support the conclusions?

Reviewer #2: Yes

3. Has the statistical analysis been performed appropriately and rigorously? 

Reviewer #2: Yes

4. Have the authors made all data underlying the findings in their manuscript fully available?

Reviewer #2: Yes

5. Is the manuscript presented in an intelligible fashion and written in standard English?

Reviewer #2: Yes

6. Review Comments to the Author

Reviewer #2: (No Response)

7. PLOS authors have the option to publish the peer review history of their article (what does this mean?). If published, this will include your full peer review and any attached files.

Reviewer #2: **Yes: **Mohamed Amine BABA
